# Temocillin: A Narrative Review of Its Clinical Reappraisal

**DOI:** 10.3390/antibiotics14090859

**Published:** 2025-08-26

**Authors:** Lavinia Cosimi, Verena Zerbato, Nina Grasselli Kmet, Alessandra Oliva, Francesco Cogliati Dezza, Nicholas Geremia, Dario Cattaneo, Kristina Nadrah, Mateja Pirs, Rajko Saletinger, Alessio Nunnari, Filippo Mearelli, Filippo Giorgio Di Girolamo, Graziana Avena, Roberta Russo, Carolina Fabiani, Sergio Venturini, Luigi Principe, Giovanna Maria Nicolò, Stefano Di Bella

**Affiliations:** 1Infectious Diseases Unit, Trieste University Hospital (ASUGI), 34125 Trieste, Italy; 2Infectious Diseases Department, University Medical Centre Ljubljana, 1000 Ljubljana, Slovenia; 3Faculty of Medicine, University of Ljubljana, 1000 Ljubljana, Slovenia; 4Department of Public Health and Infectious Diseases, Sapienza University of Rome, 00185 Rome, Italy; 5Unidad Clínica de Enfermedades Infecciosas y Microbiología, Hospital Universitario Virgen Macarena, Departamento de Medicina, Universidad de Sevilla, Instituto de Biomedicina de Sevilla (IBiS)/CSIC, 41009 Seville, Spain; 6Unit of Infectious Diseases, Department of Clinical Medicine, Ospedale “dell’Angelo”, 30174 Venice, Italy; 7Unit of Infectious Diseases, Department of Clinical Medicine, Ospedale Civile “S.S. Giovanni e Paolo”, 30122 Venice, Italy; 8Department of Biomedical Sciences, Humanitas University, Pieve Emanuele, 20072 Milan, Italy; 9Institute of Microbiology and Immunology, Faculty of Medicine, University of Ljubljana, 1000 Ljubljana, Slovenia; 10Unit of Internal Medicine, Clinica Medica, Trieste University Hospital (ASUGI), 34125 Trieste, Italy; 11Pharmacy Unit, Trieste University Hospital (ASUGI), 34125 Trieste, Italy; 12Clinical Analysis Laboratory, Trieste University Hospital (ASUGI), 34125 Trieste, Italy; 13Department of Infectious Diseases, Santa Maria Degli Angeli Hospital of Pordenone (AS FO), 33170 Pordenone, Italy; 14Microbiology and Virology Unit, Great Metropolitan Hospital “Bianchi-Melacrino-Morelli”, 89128 Reggio Calabria, Italy; 15Clinical Department of Medical, Surgical and Health Sciences, Trieste University, 34129 Trieste, Italy

**Keywords:** temocillin, ESBL, AmpC, KPC, carbapenem-sparing, antimicrobial stewardship, OPAT, *Enterobacterales*

## Abstract

**Background:** The emergence of multidrug-resistant Gram-negative bacteria, particularly extended-spectrum β-lactamase (ESBL) and AmpC-producing *Enterobacterales*, has brought renewed interest in temocillin, a narrow-spectrum β-lactam antibiotic first introduced in the 1980s. **Objectives:** We aimed to provide a comprehensive overview of the microbiological, pharmacological, and clinical profile of temocillin. **Methods:** We conducted a narrative review of the literature using the PubMed database to identify relevant studies concerning the microbiology, pharmacokinetics, pharmacodynamics, clinical applications, and safety of temocillin. **Results**: Temocillin shows strong in vitro activity against ESBL- and AmpC-producing organisms, and partial efficacy against certain *Klebsiella pneumoniae* carbapenemase (KPC)-producing strains. Its pharmacokinetic and pharmacodynamic characteristics, including β-lactamase stability and low ecological impact, support its use in urinary tract infections, bloodstream infections, intra-abdominal infections, pneumonia, and central nervous system infections. Additionally, evidence supports its utility in outpatient parenteral antimicrobial therapy (OPAT), including subcutaneous administration, and in vulnerable populations such as pediatric, elderly, and immunocompromised patients. Temocillin demonstrates a favorable safety profile, minimal disruption of gut microbiota, and cost-effectiveness. It also exhibits synergistic activity with agents like fosfomycin, further enhancing its clinical value. Most of the current evidence is derived from retrospective and observational studies. **Conclusions:** Temocillin emerges as a promising carbapenem-sparing option for the treatment of challenging infections caused by multidrug-resistant Gram-negative bacteria.

## 1. Introduction

Temocillin, a semisynthetic 6-α-methoxy derivative of ticarcillin, is a penicillin-class antibiotic first introduced in Belgium [1] and the United Kingdom during the 1980s [2]. Temocillin is currently available in some European countries (e.g., France and Germany) and in Iran [3,4,5].

Temocillin is a narrow-spectrum β-lactam antibiotic primarily active against *Enterobacterales*, including strains producing extended-spectrum β-lactamases (ESBLs) and AmpC β-lactamases, against which it demonstrates high stability and potent activity [2]. It may also retain in vitro efficacy against selected *Klebsiella pneumoniae* carbapenemase (KPC)-producing isolates, although this activity is variable and strain-dependent [6]. Beyond *Enterobacterales*, temocillin shows activity against *Haemophilus influenzae*, *Moraxella catarrhalis*, *Neisseria* species, and *Burkholderia cepacia* [7,8]. However, temocillin lacks meaningful activity against Gram-positive cocci, anaerobic bacteria, and classical non-fermenting Gram-negative bacilli like *Pseudomonas aeruginosa* and *Acinetobacter baumannii* [9].

In addition, from an antimicrobial stewardship point of view, its targeted spectrum, mainly focused on *Enterobacterales*, combined with a favorable ecological profile and low propensity to induce *Clostridioides difficile* infection, make it an attractive candidate for treating Gram-negative infections.

Pharmacokinetic and pharmacodynamic (PK/PD) studies indicate that temocillin possesses favorable properties supporting its use across a range of infections and clinical settings [10]. Clinically, it has demonstrated efficacy in the management of severe infections, in particular urinary tract infections (UTIs), bloodstream infections (BSIs), and pneumonia [2,11]. Its favorable safety profile also supports its incorporation into prolonged treatment regimens, an important consideration for patients requiring extended antimicrobial therapy [9].

This review offers a comprehensive assessment of the clinical, microbiological, and pharmacological data supporting the renewed interest in temocillin within modern medical practice.

## 2. Materials and Methods

This narrative review was conducted through a comprehensive search of the PubMed database using “temocillin” as the main search term. The search included studies published in English, focusing on microbiology, pharmacokinetics/pharmacodynamics, clinical efficacy, safety, and antimicrobial stewardship aspects of temocillin. All PubMed articles available up to July 2025 were screened, with no lower time limit applied. Articles were selected based on relevance to the scope of the review. Additional references were selected through manual screening of bibliographies to ensure completeness. No systematic review methodology or meta-analysis was applied.

## 3. Microbiology and Mechanisms of Resistance

Temocillin exerts its antimicrobial activity by binding to bacterial penicillin-binding proteins (PBPs), thereby inhibiting peptidoglycan synthesis and compromising cell wall integrity. Structurally, temocillin consists of a β-lactam ring fused to a thiazolidine ring, along with a carboxylic acid group and a distinctive 6-α-methoxy substituent [10,12]. This 6-α-methoxy group is key to its β-lactamase stability: it introduces steric hindrance that prevents water molecules from accessing the active site of serine β-lactamases, thereby inhibiting the activation of the catalytic serine and subsequent hydrolysis of the β-lactam ring [13] (Figure 1). While this structural modification provides significant protection against β-lactamase degradation, it also reduces affinity for PBPs 1, 2, and 3, narrowing the antibiotic’s spectrum of activity. However, temocillin retains high affinity for PBP5 and PBP6, which explains its selective efficacy [14,15]. Temocillin exhibits stability against a wide range of β-lactamases, including both classical and extended-spectrum enzymes such as TEM, SHV, and CTX-M variants, as well as AmpC β-lactamases [16,17]. According to the Ambler classification of β-lactamases, its stability against carbapenemases is more variable. While temocillin may retain activity against certain class A serine carbapenemases (e.g., KPC), it is generally ineffective against class D enzymes (e.g., OXA-48-like) and class B metallo-β-lactamases (e.g., IMP, VIM, NDM) [6,18]. Resistance mechanisms have also been described. Stewart et al. in 2022 observed reduced susceptibility to temocillin in *E*. *coli* strains co-producing CTX-M-134, CTX-M-55 with CMY-2, and CMY-138 [19]. Pérez-Palacios et al. in 2024 documented resistance resulting from mutations in the sensor histidine kinase BaeS, which led to overexpression of efflux pumps such as MdtABC and AcrD in *E*. *coli* and *Enterobacter cloacae* [20]. The European Committee on Antimicrobial Susceptibility Testing (EUCAST) defines the susceptibility breakpoint for temocillin as minimum inhibitory concentration (MIC) ≤ 16 mg/L for infections originating from the urinary tract caused by *E*. *coli*, *Klebsiella* spp. (except *K*. *aerogenes*), and *Proteus mirabilis* [21]. Available data on temocillin resistance are largely derived from local or regional studies. As a result, robust global epidemiological evidence is still lacking.

### 3.1. Temocillin and Pseudomonas aeruginosa

Temocillin is generally ineffective against *P*. *aeruginosa* due to structural features that impair its binding to target proteins. Although it primarily targets PBP3, the 6-α-methoxy group disrupts a key hydrogen bond, destabilizing the acyl–enzyme complex and reducing efficacy. Mutations affecting PBP binding, particularly in PBP1a/b, can further alter susceptibility and are often linked to resistance development during treatment [22,23]. The MexAB-OprM efflux pump actively reduces intracellular drug levels, and its overexpression contributes significantly to resistance. Additionally, the chromosomal AmpC β-lactamase can hydrolyze β-lactams, including temocillin to some extent, despite its general stability against ESBLs. When combined with low membrane permeability, these factors result in high-level, intrinsic resistance in *P*. *aeruginosa* [24,25,26,27]. Reduced outer membrane permeability also limits temocillin activity in *P*. *aeruginosa*. Loss or downregulation of porins like OpdK, OpdF, and especially OprD impairs drug entry, while alternative channels such as OpdP may only partially compensate [24,28].

### 3.2. Antibiotic Synergism

The synergistic effect of combining temocillin with other antibiotics against Gram-negative bacteria appears to be highly dependent on both the bacterial strain and the companion antibiotic used [29]. As a narrow-spectrum agent, temocillin has demonstrated varying degrees of synergism when paired with other antimicrobials, with the most promising results observed in combination with fosfomycin [30]. Notably, this combination has shown potent synergistic activity against KPC-producing *K*. *pneumoniae*, a multidrug-resistant pathogen of major clinical concern. According to recent findings, the temocillin–fosfomycin combination exhibited synergistic effects in 91% of tested clinical isolates and significantly improved survival outcomes in *Galleria mellonella* larvae. Furthermore, this combination was able to reduce temocillin minimum MICs below the resistance breakpoint in most otherwise resistant strains, suggesting a potential therapeutic advantage in challenging infections [30]. In contrast, the combination of temocillin with aminoglycosides, such as gentamicin, tobramycin, amikacin, and netilmicin, has generally shown limited synergistic activity. Studies report that only 12% of Gram-negative strains demonstrated synergy with the temocillin–gentamicin combination, and 7–10% with other aminoglycoside pairings [8].

## 4. Pharmacokinetics and Pharmacodynamics

Temocillin is currently approved as an intravenous (IV) or intramuscular (IM) sodium salt solution in several countries. However, emerging evidence supports its promising use via the subcutaneous (SC) route, offering a practical alternative for patients with limited venous access or those enrolled in outpatient parenteral antimicrobial therapy (OPAT) programs [31].

Alexandre and Fantin have reviewed the main pharmacokinetics characteristics of temocillin [10]. Temocillin exhibits high plasma protein binding (63–85%), a low volume of distribution (0.15–0.25 L/kg), and is primarily eliminated via the renal route, with urinary recovery ranging from 68% to 74%. Its elimination half-life varies from approximately 5 h in healthy individuals to around 30 h in patients with severe uremia [10]. Based on these properties, temocillin PKs are likely to be significantly altered in the presence of severe hypoalbuminemia, altered fluid balance (e.g., capillary leakage, edema), and/or in patients with advanced renal dysfunction undergoing dialysis. In such complex clinical scenarios, appropriate dose adjustments are required [10,32]. However, optimal dosing strategies are limited by a lack of specific PK data [10,32]. The currently licensed dosing regimen recommends reducing the dose of temocillin when creatinine clearance falls below 60 mL/min to 1 g every 12 h. If creatinine clearance is between 30 and 10 mL/min, the dose should be further reduced to 1 g every 24 h. When clearance is below 10 mL/min, a dose of 1 g or 500 mg is recommended every 24–48 h. In patients undergoing intermittent dialysis, temocillin is administered at doses of 1, 2, or 3 g on dialysis days only, depending on the duration of the inter-dialytic interval (24, 48 or 72 h) [33]. However, one study has shown that the currently licensed dosing regimen is suboptimal for pathogens with MICs > 8 mg/L [34].

Data on the tissue distribution of temocillin remain limited. However, studies conducted in both healthy volunteers and patients receiving IV administration have demonstrated satisfactory tissue penetration. Measured drug concentrations reached approximately 50–60% of corresponding serum levels in peripheral lymph nodes [35] and peritoneal fluid [36], 25–30% in pulmonary tissue [37], and around 35% in prostatic tissue [38]. Notably, temocillin concentrations in bile—assessed in patients with biliary tract diseases—were nearly 10-fold higher than the corresponding serum levels [39]. In contrast, penetration into the cerebrospinal fluid (CSF) appears limited, with reported CSF/serum concentration ratios ranging from 5% to 25%, largely depending on the presence or absence of meningeal inflammation [40,41]. The main pharmacokinetic and pharmacodynamic characteristics of temocillin are summarized in Table 1.

More recently, a single-dose PK study in healthy volunteers receiving 2 g of temocillin via IV or SC routes showed that unbound drug concentrations in muscle and SC tissues exceeded those in plasma, supporting good tissue penetration for both formulations [31].

## 5. Clinical Application and Efficacy

Temocillin has been increasingly investigated in clinical settings for the treatment of infections caused by multidrug-resistant Gram-negative bacteria. A growing number of retrospective and prospective studies, conducted since 2000 and involving cohorts of at least 10 patients, have assessed its efficacy and safety across various infection sites, including urinary tract, bloodstream, intra-abdominal, and respiratory tract infections. These findings are summarized in Table 2. Ongoing clinical trials exploring additional applications of temocillin are listed in Table 3, while its current clinical indications are outlined in Figure 2.

### 5.1. Temocillin in Urinary Tract Infections

Temocillin has been extensively evaluated for the treatment of febrile UTIs in both pediatric and adult populations [51,54]. Clinical studies have demonstrated high cure rates, including in pediatric patients with febrile UTIs caused by ESBL-producing *Enterobacteriaceae*, showing efficacy comparable to standard therapies [54].

Earlier studies reported excellent in vitro bactericidal activity against uropathogens—particularly *Enterobacterales*—with low resistance rates [7]. Recent microbiological surveillance studies have confirmed the sustained activity of temocillin, with approximately 80–90% of AmpC- or ESBL-producing Enterobacterales remaining susceptible when assessed using the EUCAST urinary-specific breakpoint (MIC ≤ 16 mg/L) [57,58].

Temocillin is well suited for the treatment of urinary tract infections due to its high urinary excretion [10]. A standard dosing regimen of 4 g/day has proven effective in non-critically ill patients. However, recent pharmacokinetic and pharmacodynamic data suggest that higher doses, such as 6 g/day, may be warranted in selected cases, particularly for the treatment of cUTIs, and are in line with updated EUCAST recommendations. In patients with moderate renal impairment, who generally exhibit reduced clearance and higher drug exposure, a standard dose of 4 g/day remains appropriate [59].

Temocillin has shown promise in the prophylaxis of recurrent ESBL-associated UTIs, as demonstrated by a recent case report, highlighting its potential role in preventive strategies [60].

### 5.2. Temocillin in Bloodstream Infections

Only some retrospective studies have demonstrated its effectiveness in BSIs caused by *Enterobacterales*. In a study by Balakrishnan et al., clinical and microbiological success rates reached 86% and 84%, respectively [2]. These findings were further supported by Dinh et al., who observed an overall treatment failure rate of 13.3% [51]. A more recent retrospective study focusing exclusively on BSIs, most of which were caused by *Escherichia coli, Klebsiella species* (excluding *Klebsiella aerogenes*), and *Proteus mirabilis,* demonstrated a clinical success rate of 92% [50].

Laterre et al., in a randomized clinical trial, found that continuous infusion of temocillin achieved higher serum concentrations compared to intermittent administration in critically ill patients, including those with BSIs [44].

The ASTARTÉ trial (NCT04478721), a completed multicenter randomized study, was designed to evaluate the non-inferiority of temocillin versus meropenem for BSIs caused by third-generation cephalosporin-resistant *Enterobacterales* [61]. The results of this trial are highly anticipated, as they may provide pivotal evidence to guide treatment decisions in this challenging clinical context.

### 5.3. Temocillin in Pneumonia

Temocillin has also been investigated as a potential treatment option for hospital-acquired pneumonia (HAP), particularly in the context of infections caused by multidrug-resistant Gram-negative organisms. In a retrospective study, Habayeb et al. compared a regimen of amoxicillin plus temocillin to piperacillin/tazobactam for the empiric treatment of severe HAP [45]. Clinical success rates were comparable between the two groups (82% vs. 80%) [45]. Kandil et al. reported a clinical success rate of 87.5% in patients with respiratory tract infections, within an overall hospitalized cohort showing 88.9% success [11]. Other studies have reported less favorable outcomes [48,55] (see Table 2 for further details).

Using population PK/PD modeling, De Jongh et al. demonstrated that continuous infusion of 6 g/day may be necessary in critically ill patients with severe pneumonia to achieve therapeutic drug concentrations both systemically and in the epithelial lining fluid. This approach may enhance efficacy and reduce the risk of resistance selection [62].

### 5.4. Temocillin in Abdominal Infections

Temocillin has shown promising clinical effectiveness in the management of intra-abdominal infections (IAIs). Clinical studies have demonstrated that adequate concentrations can be achieved at the site of infection, including peritoneal and wound secretions [63]. In one study, peritoneal levels reached 48% of serum concentrations within the first hour post-administration, with a mean of 49.1 mg/L over 3.5 h, suggesting that a dosing regimen of 1 g twice daily may be clinically sufficient for susceptible pathogens [36].

However, due to its narrow antibacterial spectrum, temocillin is not suitable for empirical treatment of polymicrobial complicated IAIs, which often involve *P*. *aeruginosa* and anaerobic bacteria [64]. Its use should therefore be limited to selected cases where such pathogens are unlikely, or in combination regimens that include anti-pseudomonal and anti-anaerobic coverage to ensure complete therapeutic adequacy.

### 5.5. Temocillin in Central Nervous System Infections

Temocillin is emerging as a potential treatment option for central nervous system (CNS) infections. Its clinical utility in this setting is closely linked to its PK profile in the CSF [65]. Recent studies have reported a median CSF penetration of approximately 12.1% in critically ill patients, which may be sufficient to achieve therapeutic concentrations against pathogens with MICs ≤ 4 mg/L. Based on these findings, a dosing regimen of 6 g/day by continuous infusion appears to be adequate for treating CNS infections caused by susceptible strains [41]. In infections caused by organisms with higher MICs, higher dosing strategies might be necessary to ensure effective CSF levels; however, the safety of such regimens still requires further investigation. Despite its favorable resistance profile and preserved activity against certain carbapenemase-producing *Enterobacterales*, temocillin is not currently recommended by major clinical guidelines for the treatment of CNS infections.

### 5.6. Temocillin in Sexually Transmitted Infections and Prostatitis

Temocillin has also shown potential in the treatment of sexually transmitted infections, particularly gonorrhea. Its remarkable stability against β-lactamases makes it active against both penicillin-sensitive and penicillinase-producing strains of *Neisseria gonorrhoeae* [66]. In two clinical studies involving a total of 267 patients with acute uncomplicated gonorrhea, single intramuscular doses of 0.5 g or 1.0 g achieved bacteriological cure rates of 91% and 94%, respectively, with excellent tolerability and minimal adverse effects [67]. Additionally, the use of temocillin in urogenital infections may extend to male accessory gland infections. A study showed that temocillin administered intravenously (2 g) reached therapeutically relevant concentrations in both central and peripheral prostatic tissues, achieving 26.1% and 34.6% of plasma levels, respectively. Importantly, these concentrations exceeded the MIC90 for most *Enterobacterales*, supporting its potential role in the treatment of prostatitis, particularly when caused by multidrug-resistant strains [38].

## 6. Use of Temocillin in Different Patient Populations

Evidence of temocillin use in special patient populations remains limited and is often derived from retrospective studies. This underscores the need for prospective, population-specific PK investigations. The following subsections summarize the available data in elderly, pediatric, pregnant, critically ill, and immunocompromised patients.

### 6.1. Elderly and Pediatric Patients

Few studies reported the use of temocillin for treating elderly patients with UTIs, IAIs, and lower respiratory tract infections. Heard et al. involved 205 patients (median age 79 years) in their study. UTIs and pneumonia were predominant. The overall clinical success rate was 79.5% with significantly better outcomes in UTIs [48]. Kandil et al. included 126 patients (median age 73 years), predominantly with intra-abdominal and lower respiratory tract infections, while fewer had UTIs. Clinical success was high (88.9%) [11]. Neither study, nor any others, identified age as an independent risk factor for treatment failure.

Temocillin has been predominantly used in the pediatric population for the treatment of UTIs, with excellent clinical cure rates [54].

### 6.2. Pregnancy and Lactation

There are no data about the safety of temocillin during pregnancy. Trace amounts of penicillin may be excreted in breast milk, and breastfeeding is therefore not recommended during treatment with temocillin [33].

### 6.3. Critically Ill Patients

The clinical efficacy of temocillin in critically ill patients varies across studies. While some studies exclusively examined critically ill patients [44], most involved patients from general wards (see Table 2 for further details).

### 6.4. Immunocompromised Patients

The efficacy of temocillin in immunocompromised patients has been demonstrated in few studies. Delory et al. reported a 94% clinical cure rate in 72 patients with ESBL-positive UTIs, 56% of whom were immunocompromised (primarily kidney transplant recipients), with no significant differences based on immune status [46]. Dinh observed an overall treatment success rate of 86.7% among patients with various infections, 54% of whom were immunocompromised. Multivariate analysis showed that immunocompromised status was not a risk factor for treatment failure [51]. However, Oosterbos reported a 20% clinical failure rate in immunocompromised patients with bacteremia, significantly higher than the rest of the cohort. Most cases of clinical failure occurred in patients with intra-abdominal or lower respiratory tract infections as the source of bacteremia [50].

### 6.5. Evaluating Temocillin for OPAT

Temocillin solutions maintained chemical stability—defined as ≥90% of the initial concentration—for at least 24 h at 37 °C, and for at least 72 h at lower temperatures (at 4 °C, 25 °C, and 32 °C), both in polypropylene infusion bags and in polyisoprene elastomeric pumps [68]. Accordingly, temocillin can reasonably be considered a suitable candidate for OPAT, particularly in patients eligible for early discharge and continued treatment in an outpatient setting. A study evaluating 57 OPAT episodes, primarily involving UTIs, reported a late clinical cure rate of 85.7%, with improved outcomes associated with non-disseminated infections and MIC values ≤ 8 mg/L [53]. These data highlight temocillin’s growing role in OPAT settings.

### 6.6. Subcutaneous Administration of Temocillin

The SC administration of temocillin offers a potentially practical alternative to the IV route, particularly in outpatient settings, though it is not limited to them. In a randomized two-period crossover study, Matzneller et al. investigated the single-dose pharmacokinetics of temocillin following both IV and SC administration in healthy volunteers [31]. The study showed that IV dosing led to a higher C max (233.5 ± 50.2 mg/L vs. 100.0 ± 15.5 mg/L, *p* = 0.0001), but SC infusion provided more sustained plasma concentrations over time, with higher levels observed at 12 h post-dose (50.3 ± 15.3 mg/L vs. 29.9 ± 7.4 mg/L, *p* = 0.002). The relative bioavailability of SC administration was estimated at 112% ± 19%. Notably, unbound temocillin exposure in soft tissues was substantial, with an area under the plasma concentration–time curve from 0 to 12 h (AUC_0–12_) of 105.7 ± 29.6 mg·h/L in muscle and 173.4 ± 53.1 mg·h/L in *subcutis*, exceeding unbound plasma levels (67.2 ± 13.3 mg·h/L). This translated into favorable fT > MIC values in *subcutis* (74.5% for MIC = 8 mg/L), suggesting effective tissue penetration. While some participants reported mild to moderate local infusion site discomfort (e.g., burning, pain), both routes were generally well tolerated [31]. These findings support the feasibility and pharmacological soundness of SC temocillin, particularly in patients with difficult IV access or requiring continuous parenteral therapy in home or palliative care settings, although its broader use remains limited by scarce clinical experience and the need for further validation.

## 7. Safety and Tolerability

Both preclinical and clinical safety evaluations have consistently supported the favorable risk–benefit profile of temocillin. Toxicological studies, including acute, subacute, and chronic assessments in animal models, as well as reproductive toxicity and mutagenicity evaluations, have demonstrated good tolerability even at high doses, with no significant safety concerns at therapeutic levels [69]. The most reported adverse events are gastrointestinal, with diarrhea being the most common manifestation [45,56]. Notably, cases of diarrhea caused by *C*. *difficile* have been reported; however, these remain infrequent and are typically mild [46,51]. The second most frequent adverse event involves skin reactions, predominantly maculopapular rashes [51]. However, a case of vascular *purpura* has also been reported, though it was not definitively linked to temocillin use [56]. Rare cases of acute kidney injury (AKI) have been described during temocillin therapy [51,56].

## 8. Cost Analysis and Economic Considerations

A key advantage of temocillin is its cost-effectiveness. In Sweden, a cost-effectiveness analysis demonstrated that temocillin resulted in better health outcomes compared to cefotaxime for febrile UTIs. The incremental cost-effectiveness ratio (ICER) was EUR 38,400 per quality-adjusted life-year gained. Furthermore, temocillin had a 63% probability of being cost-effective at a willingness-to-pay threshold of EUR 50,000, emphasizing the importance of incorporating antibiotic resistance into economic evaluations to optimize healthcare resources [70]. A budget impact analysis conducted in Iran further supported temocillin’s economic value. The study revealed that including temocillin in insurance coverage for treating UTIs caused by ESBL-producing bacteria could reduce overall treatment costs from USD 36 million to USD 34 million, resulting in annual savings exceeding USD 9 million. These savings stem from reduced expenses related to the management of antimicrobial resistance and hospital-acquired infections [5]. Despite these promising findings, broader research is essential to validate temocillin’s cost-effectiveness across diverse healthcare systems. Differences in healthcare costs, resistance patterns, and antibiotic accessibility necessitate localized economic evaluations to ensure the generalizability of these results. Integrating temocillin into standard treatment protocols could further amplify its benefits, reducing dependence on carbapenems and enhancing global antimicrobial stewardship initiatives.

## 9. Conclusions and Future Directions

The accelerating global spread of ESBL- and AmpC-producing *Enterobacterales* represents a major challenge to modern infectious disease management and underscores the urgent need for targeted, sustainable antimicrobial strategies. In this context, temocillin is re-emerging as a valuable therapeutic option due to its high stability against β-lactamases, favorable safety profile, and minimal ecological disruption. Although clinical data support the use of temocillin in selected infections, particularly urinary tract infections caused by multidrug-resistant *Enterobacterales*, its broader application remains limited by the scarcity of evidence in severe, systemic, or non-urinary infections. Future research should aim to define temocillin’s full clinical potential through well-designed studies that explore its efficacy in diverse infection sites, its role in combination therapy, and its performance under optimized PK/PD regimens through continued surveillance of emerging resistance. With growing evidence, temocillin may emerge as a useful component of rational antimicrobial use and a promising alternative to carbapenems for the treatment of serious infections caused by resistant Gram-negative pathogens.

## Figures and Tables

**Figure 1 antibiotics-14-00859-f001:**
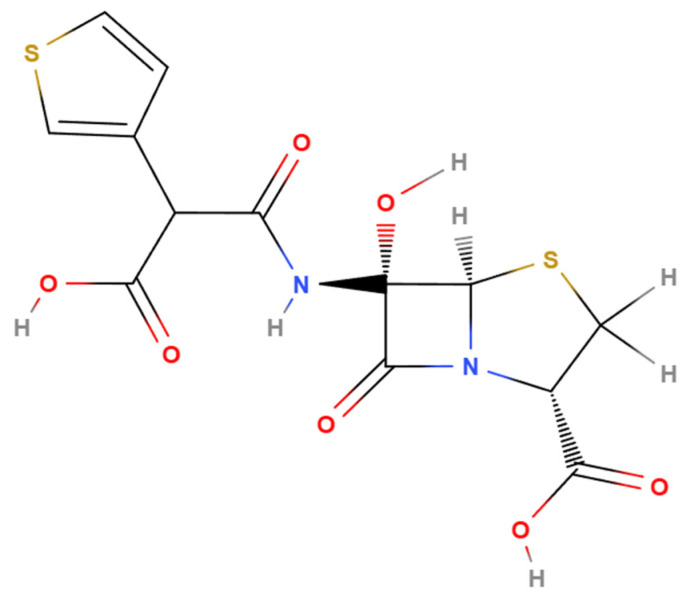
Chemical structure of temocillin (generated with MolView (v2.4), accessed 9 July 2025; available at https://molview.org/).

**Figure 2 antibiotics-14-00859-f002:**
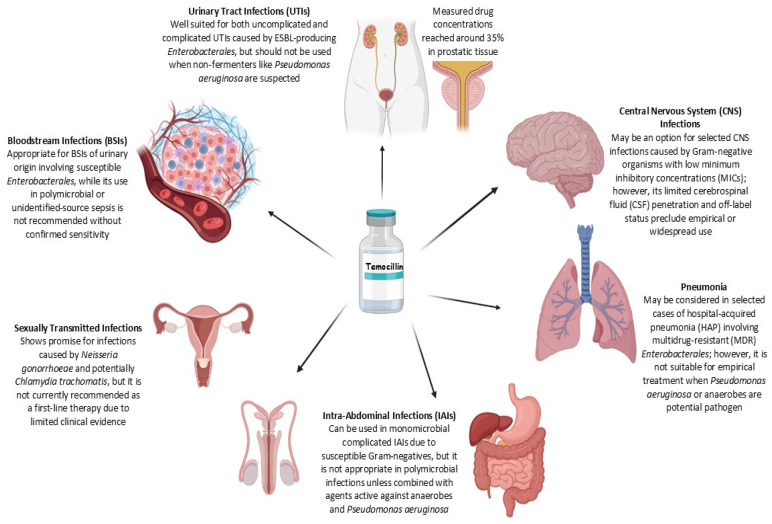
Summary of temocillin use across infection types. The figure outlines the clinical applicability of temocillin in various infectious disease settings (generated with BioRender.com, accessed 9 July 2025; available at https://biorender.com/).

**Table 1 antibiotics-14-00859-t001:** Summary of PK/PD data.

Parameter	Value/Range
Protein binding	63–85%
Volume of distribution (Vd)	0.15–0.25 L/kg
Elimination route	Renal (urinary recovery 68–74%)
Elimination half-life	≈5 h
Tissue penetration (peritoneal fluid)	≈50–60% of serum levels
Tissue penetration (lung)	≈25–30% of serum levels
Tissue penetration (prostate)	≈35% of serum levels
CSF penetration	5–25% of serum

Cerebrospinal fluid: CSF.

**Table 2 antibiotics-14-00859-t002:** Summary of clinical studies evaluating the efficacy of temocillin in various infection types.

Author and Year	Study Design	Clinical Indication	Patients Treated with Temocillin	Temocillin Dosing Regimen	Outcomes	Reference
(Lekkas et al., 2006)	Retrospective, monocentric	Acute exacerbations in CF caused by *Burkholderia cepacia complex*	23 (36 episodes)	Mean dose: 4 g/day (range 2–6) Mean duration: 14 days (range 1–40).	Clinical improvement: 56.25% (18/32)	[42]
(Kent et al., 2008)	Retrospective, multicentric (temocillin vs. comparator)	Acute exacerbations in CF mainly caused by *Pseudomonas aeruginosa* e *Burkholderia cepacia*	26	N/A	Improvement in FEV1% was similar for the temocillin and non-temocillin courses	[43]
(Balakrishnan et al., 2011)	Retrospective, multicentric	UTI, BSI and HAP	92	4 g/day	Clinical cure: 86% (79/92) Microbiological cure: 84% (66/79)	[2]
(Laterre et al., 2015)	RCT (continuous vs. intermittent administration)	Critical patients with UTI, IAI, LRTI and BSI	32	6 g/day	Continuous infusion achieved higher serum concentrations than intermittent dosing.	[44]
(Habayeb et al., 2015)	Retrospective, monocentric (temocillin + amoxicillin vs. piperacillin/tazobactam)	Severe HAP	94 (98 episodes)	4 g/day Mean duration: 6.8 ± 1.5 days	Clinical cure: 82% (80/94) *C*. *difficile* episodes: 4% (4/98)	[45]
(Delory et al., 2021)	Retrospective, multicentric (temocillin vs. carbapenems)	UTI caused by ESBL-producing *Enterobacterales*	72	Median dose: 4 g/day (IQR 2–4) Median duration: 14 days (IQR 12–18)	Clinical cure: 94% (68/72)	[46]
(Alexandre et al., 2021)	Retrospective, monocentric	Mainly UTI caused by ESBL-producing *Enterobacterales*	153	78.4%: 2 g q12 h 17.6%: 2 g q8 h 3.9%: 21 g q12 h Median duration: 14 days (IQR 7–17)	Early clinical failure (UTI): 4.9% (6/123) Early clinical failure (non-UTI) 26.7% (8/30) 30-day mortality (UTI): 1.6% (2/123) 30-day mortality (non-UTI): 13.3% (4/30)	[47]
(Heard et al., 2021)	Retrospective, monocentric	Mainly UTI and LRTI	205	4 g/day Median duration: 5.9 days (4.6–7.8)	Clinical success: 79.5% (163/205) Higher success in UTI (85.8%) vs. LRTI (67.9%)	[48]
(Edlund et al., 2022)	RCT phase 4 (temocillin vs. cefotaxime)	Febrile UTI	77	2 g q8 h	Temocillin less disruptive to microbiota than cefotaxime	[49]
(Oosterbos et al., 2022)	Retrospective, monocentric	BSI mainly caused by *Escherichia coli, Klebsiella* species (except *Klebsiella aerogenes*) and *Proteus mirabilis*	172 (182 episodes)	39%: 2 g q12 h 61%: 2 g q8 h	Clinical success: 92% (167/182)	[50]
(Dinh et al., 2022)	Retrospective, multicentric	Mainly UTI caused by ESBL-producing *Enterobacterales*	113	Mean dosage: 5.4 ± 1.5 g per day Mean duration: 9.2 ± 6.9 days	Clinical cure: 86.7% (98/113)	[51]
(Enoch et al., 2022)	Retrospective, monocentric	Mainly UTI caused by ESBL-producing *Enterobacterales*	24	4 g/day Mean duration: 6 days	Recurrence: 8%	[52]
(Van den Broucke et al., 2022)	Prospective, monocentric	Mainly UTI	50 (57 episodes)	4 g/day (OPAT)	Clinical cure: 85.7% (48/56)	[53]
(Kandil et al., 2023)	Retrospective, monocentric	Mainly IAI, LRTI and UTI	126	4 g/day Median duration: 5 days (IQR 4–7)	Clinical success: 88.9% (112/126)	[11]
(Bayart et al., 2024)	Retrospective, monocentric (temocillin vs. comparator)	Children with febrile UTI caused by ESBL-producing *Enterobacterales*	36	Median duration: 7 days (IQR 7–9)	Clinical cure: 100% (36/36)	[54]
(Mamona Kilu et al., 2024)	Retrospective, multicentric	Mainly LRTI and IAI.; 82% caused by ESBL-producing *Enterobacterales*	163	Variable dose: ≥6 g/day: 63.6% 4 g/day: 27.1% <4 g/day: 9.3% Median duration: 7 days (IQR 4–11)	Clinical failure: 28.1% (36/128)	[55]
(Lahouati et al., 2024)	Retrospective, monocentric	BJI caused by ESBL-producing *Enterobacterales*	17	Median dose: 6 g/day for 42 days (IQR 14–42)	Clinical cure: 66.7% (8/12)	[56]
(Brousse et al., 2025)	Retrospective, multicentric	cUTI and BJI caused by AmpC β-lactamase-producing *Enterobacterales*	67	Mean dose: 4.2 g/day range (0.5–6) Mean duration: 13 days (range 1–45)	Clinical success: 89% (56/63)	[57]

BSI: bloodstream infection; CF: cystic fibrosis; cUTI: complicated urinary tract infection; BJI: bone and joint infection; LRTI: lower respiratory tract infection; IAI: intra-abdominal infection; IQR: interquartile range; N/A: not applicable; OPAT: outpatient parenteral antimicrobial therapy; RCT: randomized controlled trial.

**Table 3 antibiotics-14-00859-t003:** Summary of ongoing clinical trials investigating temocillin (source: ClinicalTrials.gov; accessed 20 July 2025).

NCT Number	Study Title	Status	Main Outcome Assessed	Sponsor	Study Type
NCT02285075	Temocillin Pharmacokinetic in Hemodialysis	COMPLETED	Target attainment of temocillin in patients under hemodialysis	AZ Sint-Jan AV	INTERVENTIONAL—open label
NCT02260102	Temocillin Pharmacokinetics in Paediatrics (TEMOPEDI)	UNKNOWN STATUS	Target attainment of temocillin in children with UTI, cholangitis or requiring antibiotic prophylaxis following a hepatic transplant	Université Catholique de Louvain	INTERVENTIONAL—open label, non-randomized
NCT05413772	Temocillin in ESBL-Enterobacteriaceae Infections (TMO2016)	COMPLETED	Efficacy of temocillin in ESBL infections	Assistance Publique—Hôpitaux de Paris	OBSERVATIONAL
NCT03557840	Plasma Protein Binding and PK/PD of Total and Unbound Temocillin Non-ICU Patients (TEMODELTA)	UNKNOWN STATUS	Target attainment of temocillin in non-ICU Patients with bacterial infections (UTI, IAI, LRTI)	Paul M. Tulkens	INTERVENTIONAL—open label
NCT03599999	Study of Adverse Effects Occuring in Patients Receiving an Antibiotic Treatment with Temocillin	COMPLETED	Safety of temocillin in ESBL infections	Centre Hospitalier Universitaire, Amiens	OBSERVATIONAL
NCT04671290	Temocillin Versus Carbapenems for Urinary Tract Infection Due to ESBL-producing Enterobacteriaceae (TEMO-BLSE)	COMPLETED	Efficacy of temocillin compared to carbapenems for ESBL-E UTI	Centre Hospitalier Annecy Genevois	OBSERVATIONAL
NCT02681263	Efficacy of Temocillin in Urinary Tract Infection Due to ESBL Producing and AmpC Hyperproducing Enterobacteriaceae (TEMO-ESBL)	UNKNOWN STATUS	Efficacy of temocillin in ESBL and AmpC Enterobacteriaceae infections	University Hospital, Grenoble	INTERVENTIONAL—open label
NCT04478721	Temocillin vs. Meropenem for the Targeted Treatment of Bacteraemia Resistant to Third Gen Cephalosporins (ASTARTÉ)	COMPLETED	Non inferiority of temocillin versus carbapenems for the treatment of ESBL-producing Enterobacteriaceae bacteremia	Fundación Pública Andaluza para la gestión de la Investigación en Sevilla	INTERVENTIONAL—phase 3 RCT
NCT03543436	Temocillin Versus a Carbapenem as Initial Intravenous Treatment for ESBL Related Urinary Tract Infections (TEMO-CARB)	COMPLETED	Non inferiority of temocillin versus carbapenems for the treatment of ESBL-producing Enterobacteriaceae UTI	Assistance Publique—Hôpitaux de Paris	INTERVENTIONAL—phase 3 RCT
NCT02959957	Disturbance of the Intestinal Microbiota by Temocillin vs. Cefotaxime in Treatment of Febrile Urinary Tract Infections	COMPLETED	Safety of temocillin compared to cefotaxime in febrile UTI	Håkan Hanberger	INTERVENTIONAL—open label, randomized
NCT01543347	Temocillin Use in Complicated Urinary Tract Infections Due to Extended Spectrum Beta-Lactamases (ESBL)/AmpC Enterobacteriaceae (TEA)	WITHDRAWN	Efficacy of temocillin in ESBL and AmpC Enterobacteriaceae cUTI	Belpharma s.a.	INTERVENTIONAL—open label
NCT05565222	Piperacillin-tazobactam and Temocillin as Carbapenem-alternatives for the Treatment of Severe Infections Due to Extended-spectrum Beta-lactamase-Producing Gram-negative Enterobacteriaceae in the Intensive Care Unit (PITAGORE)	RECRUITING	Comparison of temocillin or piperacillin-tazobactam versus carbapenems for the treatment of severe ESBL-producing Enterobacterales infections in ICU patients	Assistance Publique—Hôpitaux de Paris	INTERVENTIONAL—phase 3 RCT
NCT04436991	Antibiotic Dosing in Geriatric Patients At the Emergency Department	RECRUITING	Target attainment of beta-lactams (including temocillin) in frail geriatric inpatients	University Hospital, Ghent	OBSERVATIONAL
NCT03440216	Population Pharmacokinetics and Pharmacodynamics of Beta-lactams of Interest in Adult Patients From Intensive Care Units (Pop-PK/PD)	UNKNOWN STATUS	Target attainment of beta-lactams (including temocillin) in ICU patients with bacterial infections	Université Catholique de Louvain	INTERVENTIONAL—open label, non-randomized
NCT07070102	Population Pharmacokinetics of Temocillin in Acute Enterobacterial Pyelonephritis in Children (TEMOKID-POP)	NOT YET RECRUITING	Target attainment of temocillin in children with febrile Gram-negative UTI	Assistance Publique—Hôpitaux de Paris	OBSERVATIONAL

cUTI: complicated urinary tract infection; ESBL: extended-spectrum beta-lactamase; ESBL-E: extended-spectrum beta-lactamase-producing Enterobacterales; ICU: intensive care unit; IAI: intra-abdominal infection; LRTI: lower respiratory tract infection; RCT: randomized controlled trial; UTI: urinary tract infection.

## Data Availability

No new data were created or analyzed in this study. Data sharing is not applicable to this article.

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
