# Peer review of "Temocillin: A Narrative Review of Its Clinical Reappraisal"

_antibiotics, 2025, doi:10.3390/antibiotics14090859_

Round 1
Reviewer 1 Report
Comments and Suggestions for Authors
This manuscript provides a comprehensive and up-to-date review of temocillin, focusing on its pharmacokinetics, pharmacodynamics, antimicrobial spectrum, clinical applications, and role in antimicrobial stewardship. The topic is highly relevant given the increasing demand for carbapenem-sparing agents in light of the rising prevalence of multidrug-resistant Gram-negative infections. The authors have compiled extensive clinical and pharmacological data and structured the manuscript effectively; however, several issues related to grammar, formatting, citation consistency, and scientific clarity need to be addressed to improve readability and ensure compliance with journal standards
Please address the following specific comments
- Throughout the manuscript, there are unnecessary hyphenations that should be removed, particularly from the abstract to the conclusion. For example, see lines 40, 41, 191, and 192
- Line 51: "temocillin" should be capitalized to "Temocillin"
- Line 51: Consider removing “PK/PD” or elaborating it for clarity as “pharmacokinetics/pharmacodynamics”
- Line 58: Reference formatting is inconsistent, e.g., “[3][4][5]”. Please revise to “[3–5]” as per journal style
- Line 108: Citation format should be corrected to match the journal’s referencing style
- Line 130: The citation formatting “[24][25][26][27]” needs to be corrected
- Line 143: Specify the type of experimental models mentioned in “significantly improved survival outcomes in experimental models” and add appropriate citations
- Figure 2: Please provide the raw file and indicate which software was used to create the figure, such as BioRender or PowerPoint
- Line 174: The comma after “[34]” is grammatically incorrect and should be removed
- Lines 222, 224, 228, 242: Ensure citation formatting aligns with journal guidelines
- Lines 294 and 296: Remove the space before the percent symbol
- Table 2: Correct “NOT YET RECRUTING” to “NOT YET RECRUITING”
- Format all abbreviations consistently and ensure each is defined upon first mention in the manuscript
Author Response
This manuscript provides a comprehensive and up-to-date review of temocillin, focusing on its pharmacokinetics, pharmacodynamics, antimicrobial spectrum, clinical applications, and role in antimicrobial stewardship. The topic is highly relevant given the increasing demand for carbapenem-sparing agents in light of the rising prevalence of multidrug-resistant Gram-negative infections. The authors have compiled extensive clinical and pharmacological data and structured the manuscript effectively; however, several issues related to grammar, formatting, citation consistency, and scientific clarity need to be addressed to improve readability and ensure compliance with journal standards.
Please address the following specific comments
Throughout the manuscript, there are unnecessary hyphenations that should be removed, particularly from the abstract to the conclusion. For example, see lines 40, 41, 191, and 192
We thank the reviewer for the suggestion. We have removed the unnecessary hyphenations.
Line 51: "temocillin" should be capitalized to "Temocillin"
We changed as suggested.
Line 51: Consider removing “PK/PD” or elaborating it for clarity as “pharmacokinetics/pharmacodynamics”
We have removed the key word.
Line 58: Reference formatting is inconsistent, e.g., “[3][4][5]”. Please revise to “[3–5]” as per journal style
We have corrected all the references in the text in accordance with the journal’s guidelines.
Line 108: Citation format should be corrected to match the journal’s referencing style
We have corrected all the references in the text in accordance with the journal’s guidelines.
Line 130: The citation formatting “[24][25][26][27]” needs to be corrected
We have corrected all the references in the text in accordance with the journal’s guidelines.
Line 143: Specify the type of experimental models mentioned in “significantly improved survival outcomes in experimental models” and add appropriate citations
We have specified the type of experimental model as suggested.
Figure 2: Please provide the raw file and indicate which software was used to create the figure, such as BioRender or PowerPoint
We thank the reviewer for the comment. We have specified in the manuscript that Figure 2 was created using BioRender. Since this figure is a schematic illustration and not based on experimental raw data, we believe that providing the original editable file is not required according to the journal’s guidelines.
Line 174: The comma after “[34]” is grammatically incorrect and should be removed
We have corrected the mistake.
Lines 222, 224, 228, 242: Ensure citation formatting aligns with journal guidelines
We have changed the text as suggested.
Lines 294 and 296: Remove the space before the percent symbol
We have changed the text as suggested.
Table 2: Correct “NOT YET RECRUTING” to “NOT YET RECRUITING”
We have corrected the mistake.
Format all abbreviations consistently and ensure each is defined upon first mention in the manuscript
We have checked and every abbreviation is defined upon first mention in the text.
Reviewer 2 Report
Comments and Suggestions for Authors
Comments and Suggestions for the Authors
The review article, “Temocillin: A Narrative Review of Its Clinical Reappraisal' presents an interesting reading
experience and well framed. In my opinion, at current state, the article can be accepted for publishing with a few
Minor revisions.
The following are my comments and suggestions to improve the quality of the manuscript,
• The title is sufficiently complete.
• Abstract is good.
• The introduction is concise.
• Authors should consider grouping continuous reference numbers like (24-27) instead of all individual
numbers for improved readability. There are a few more examples present in the article.
• Tables are impressive and complete.
• Line 385, authors should consider clarifying "minimal ecological disruption".
• Claims made by the authors in the article (line 391-394) that temocillin could become a “cornerstone”, is little
ambitious. Authors should also consider letting readers know about the pitfalls/limitations or potential
roadblocks that could arise like- need for regulatory approval.
• References are okay.
Author Response
Comments and Suggestions for the Authors
The review article, “Temocillin: A Narrative Review of Its Clinical Reappraisal' presents an interesting reading experience and well framed. In my opinion, at current state, the article can be accepted for publishing with a few Minor revisions.
The following are my comments and suggestions to improve the quality of the manuscript,
- The title is sufficiently complete.
- Abstract is good.
- The introduction is concise.
- Authors should consider grouping continuous reference numbers like (24-27) instead of all individual numbers for improved readability. There are a few more examples present in the article.
We have corrected all the references in the text in accordance with the journal’s guidelines.
- Tables are impressive and complete.
- Line 385, authors should consider clarifying "minimal ecological disruption".
We believe that this concept is well clarified in the introduction.
- Claims made by the authors in the article (line 391-394) that temocillin could become a “cornerstone”, is little ambitious. Authors should also consider letting readers know about the pitfalls/limitations or potential roadblocks that could arise like- need for regulatory approval.
We thank the reviewer for this thoughtful comment. We agree that describing temocillin as a potential “cornerstone” of antimicrobial therapy may sound ambitious without acknowledging existing limitations. We have therefore revised the conclusion.
- References are okay.
Reviewer 3 Report
Comments and Suggestions for Authors
This is a comprehensive narrative review on the reappraisal of temocillin, an underutilized β-lactam antibiotic with renewed relevance in an era of growing ESBL- and AmpC-mediated resistance. The manuscript is appropriately structured, covers a broad clinical and pharmacological landscape, and presents important findings supported by both literature and ongoing clinical trials. The writing is clear and informative, and the extensive bibliography adds much depth to the overall article.
There are several improvements that I would like to suggest to better present this article:
- Additional details for the “Materials and Methods” section. Although it is a narrative review, a basic outline of the following would greatly enhance reproducibility and credibility:
- Search terms
- Timeframe
- Inclusion/exclusion criteria
- Delineation of Evidence Strength:
- Currently, studies of varying quality (case reports, retrospective cohorts, RCTs) are intermingled without clear hierarchy.
Suggestion:
- Use evidence grading (e.g., RCT > cohort > case series) or clearly indicate the study type.
- Consider a “level of evidence” column in Table 1.
- Adding a small section dedicated to Critical Appraisal
- While this review is largely descriptive. A critical appraisal of key findings, especially from observational studies, would be very helpful for readers.
Suggestion: Critically appraise said studies on the following areas
- Limitations of retrospective designs
- Sample sizes
- Confounding factors
- Heterogeneity in dosing regimens
- Clinically relevant integration of PK/PD:
- The PK/PD section is strong but overly technical in places. It would benefit from:
- Practical dosing recommendations for specific infection types
- Considerations for special populations (e.g., obesity, dialysis)
Suggestion: A concise summary table of recommended doses per indication and renal function would be useful.
5. Advise caution for Subcutaneous Administration
- The SC section is promising, but the risks (local reactions, variability in absorption) could be better emphasized.
Suggestion: Explicitly mention that this is still an emerging, off-label practice, and more data is required.
6. Additional points for the Conclusion Section
- Several details could be added to better summarize future direction of researches concerning Temocillin
Suggestion: Add a bullet list or paragraph highlighting:
- Currently unanswered questions
- Research priorities (e.g. combination therapy trials, dosing optimization)
- Surveillance needs for resistance
Minor Comments:
ï‚· Line 41: Suggest replacing “intra-abdominal infections” with “complicated intra-abdominal infections (cIAIs)” for clarity.
ï‚· Line 291: Replace “Heard’s study” with a full citation format for consistency.
ï‚· Consider harmonizing the terminology for study types (e.g., avoid interchanging “RCT” with “randomized study” unless specific).
ï‚· Ensure all figures are properly labelled and referenced in the main text. Currently, Figure 1 appears without inline discussion.
ï‚· Table 1: Add a column for “Infection type(s)” to make it easier to filter data by clinical relevance.
Author Response
This is a comprehensive narrative review on the reappraisal of temocillin, an underutilized β-lactam antibiotic with renewed relevance in an era of growing ESBL- and AmpC-mediated resistance. The manuscript is appropriately structured, covers a broad clinical and pharmacological landscape, and presents important findings supported by both literature and ongoing clinical trials. The writing is clear and informative, and the extensive bibliography adds much depth to the overall article.
There are several improvements that I would like to suggest to better present this article:
- Additional details for the “Materials and Methods” section. Although it is a narrative review, a basic outline of the following would greatly enhance reproducibility and credibility:
Search terms
Timeframe
Inclusion/exclusion criteria
We thank the reviewer for this suggestion. We have made the necessary revisions to the paragraph.
- Delineation of Evidence Strength:
Currently, studies of varying quality (case reports, retrospective cohorts, RCTs) are intermingled without clear hierarchy.
Suggestion:
Use evidence grading (e.g., RCT > cohort > case series) or clearly indicate the study type.
Consider a “level of evidence” column in Table 1.
We thank the reviewer for this suggestion. We would like to point out that Table 1 (now 2) already includes a dedicated column for “study design,” which provides readers with a clear indication of the type of evidence for each study. We believe this adequately addresses the reviewer’s concern without the need for an additional “level of evidence” column.
- Adding a small section dedicated to Critical Appraisal
While this review is largely descriptive. A critical appraisal of key findings, especially from observational studies, would be very helpful for readers.
Suggestion: Critically appraise said studies on the following areas
Limitations of retrospective designs
Sample sizes
Confounding factors
Heterogeneity in dosing regimens
We thank the reviewer for this comment. We fully agree on the importance of critically appraising the available evidence. To this end, we have structured Table 2 to specifically highlight the key limitations of the included studies, such as retrospective design, sample size, and heterogeneity in dosing regimens. We believe that this structured presentation already provides the critical appraisal requested, and therefore we have opted not to add a separate section, in order to avoid redundancy.
- Clinically relevant integration of PK/PD:
The PK/PD section is strong but overly technical in places. It would benefit from:
Practical dosing recommendations for specific infection types
Considerations for special populations (e.g., obesity, dialysis)
Suggestion: A concise summary table of recommended doses per indication and renal function would be useful.
We thank the reviewer for this comment. In the PK/PD section we already discuss dosing considerations (including patients on dialysis), and in Section 6 we specifically address the use of temocillin in special patient populations. To our knowledge, PK/PD data in obese patients are currently lacking.
- Advise caution for Subcutaneous Administration
The SC section is promising, but the risks (local reactions, variability in absorption) could be better emphasized.
Suggestion: Explicitly mention that this is still an emerging, off-label practice, and more data is required.
We thank the reviewer for this important comment. We agree that the SC route, although promising, should be presented with appropriate caution. We have therefore revised the paragraph.
- Additional points for the Conclusion Section
Several details could be added to better summarize future direction of researches concerning Temocillin
Suggestion: Add a bullet list or paragraph highlighting:
Currently unanswered questions
Research priorities (e.g. combination therapy trials, dosing optimization)
Surveillance needs for resistance
We thank the reviewer for this useful comment. We agree on the importance of surveillance, and we have added this point to the conclusion. The other aspects suggested (combination therapy, dosing optimization, unanswered clinical questions) are already addressed in the conclusion section.
Minor Comments:
Line 41: Suggest replacing “intra-abdominal infections” with “complicated intra-abdominal infections (cIAIs)” for clarity.
We thank the reviewer for this suggestion. However, in this context we believe that the broader term “intra-abdominal infections” is more appropriate, as it encompasses both complicated and uncomplicated forms. For this reason, we would prefer to retain the original wording.
Line 291: Replace “Heard’s study” with a full citation format for consistency.
We changed the text as suggested.
Consider harmonizing the terminology for study types (e.g., avoid interchanging “RCT” with “randomized study” unless specific).
We thank the reviewer for the suggestion.
Ensure all figures are properly labelled and referenced in the main text. Currently, Figure 1 appears without inline discussion.
We thank the reviewer for this comment. Figure 1 is in fact cited in the main text at line 98.
Table 1: Add a column for “Infection type(s)” to make it easier to filter data by clinical relevance.
We thank the reviewer for this suggestion. We would like to point out that Table 2 (previous 1) already includes a dedicated column titled “Clinical indication,” which specifies the type(s) of infection for each study. We believe this adequately addresses the reviewer’s concern.
Reviewer 4 Report
Comments and Suggestions for Authors
Abstract: The abstract lacks specific data or quantitative summaries (e.g., success rates). Include 1–2 specific numerical outcomes in the abstract (e.g., clinical success rates).
Introduction: Lacks global epidemiological data or resistance prevalence to demonstrate the significance.
Methodology: No keywords, search strategy, time frame, or study selection process are described. Provide more transparency on search terms, selection criteria, and data extraction process even if it is a narrative review. A PRISMA-style flowchart, would enhance rigor.
Microbiology and Mechanisms of Resistance: Summarize key resistance mechanisms in a schematic diagram or table.
Pharmacokinetics and Pharmacodynamics: Add a PK/PD summary table including %fT>MIC, half-life, Vd, protein binding, and CSF penetration. Also, highlight limitations of SC data.
Clinical Applications:
a. Intra-abdominal and CNS infection sections lack strong clinical evidence but still make broad claims.
b. there seems to be no evaluation of evidence quality. Add a short summary paragraph on evidence gaps or GRADE-like rating per indication.
Use of temocillin in different patient population: Elderly sections mostly rely on retrospective data without statistical analysis. Elaborate the need for prospective, population-specific PK studies. Also, suggest caution in extrapolating dosing without pharmacometric validation in these groups.
Safety and Tolerability: Add a table summarizing adverse events across studies with frequency and severity.
Cost Analysis and Economic Considerations: Very narrow geographic scope (Sweden and Iran only) have been used. Assumptions underlying cost-savings are not critically analyzed.
Conclusion and Future Directions: The conclusion reiterates prior content and lacks a concrete future roadmap. Propose future research domains
Author Response
Abstract: The abstract lacks specific data or quantitative summaries (e.g., success rates). Include 1–2 specific numerical outcomes in the abstract (e.g., clinical success rates).
We appreciate the reviewer’s suggestion to include specific numerical outcomes in the abstract. However, given the heterogeneity of the available studies and the narrative nature of our review, we believe that providing precise success rates or quantitative summaries might be misleading. Instead, with Table 2 we aimed to offer a comprehensive qualitative synthesis that captures the overall trends and key messages emerging from the literature, which we feel is more appropriate in this context.
Introduction: Lacks global epidemiological data or resistance prevalence to demonstrate the significance.
We thank the reviewer for this insightful comment. We fully agree that global epidemiological data would strengthen the introduction. However, to date, available studies on temocillin resistance are heterogeneous, and largely restricted to local or regional settings. To our knowledge, no comprehensive global surveillance data are available. We have therefore clarified this limitation in the third paragraph of the manuscript.
Methodology: No keywords, search strategy, time frame, or study selection process are described. Provide more transparency on search terms, selection criteria, and data extraction process even if it is a narrative review. A PRISMA-style flowchart, would enhance rigor.
We thank the reviewer for this suggestion. We have made the necessary revisions to the paragraph.
Microbiology and Mechanisms of Resistance: Summarize key resistance mechanisms in a schematic diagram or table.
We thank the reviewer for this suggestion. The key resistance mechanisms are already described within the text, and we feel that an additional schematic or table would largely duplicate this information without adding substantial clarity. For this reason, we opted to maintain the narrative format, which we believe ensures conciseness and readability.
Pharmacokinetics and Pharmacodynamics: Add a PK/PD summary table including %fT>MIC, half-life, Vd, protein binding, and CSF penetration. Also, highlight limitations of SC data.
We have highlighted the limitations of SC data. We have added a table with PK/PD data of temocillin.
Clinical Applications:
- Intra-abdominal and CNS infection sections lack strong clinical evidence but still make broad claims.
We thank the reviewer for this observation. We agree that the available evidence for intra-abdominal and CNS infections is limited. In these sections we already aimed to convey a cautious message,
- there seems to be no evaluation of evidence quality. Add a short summary paragraph on evidence gaps or GRADE-like rating per indication.
We thank the reviewer for this suggestion. We agree that highlighting the quality of available evidence is important. To this end, Table 2 already summarizes all published studies and specifies their design, thereby allowing readers to appreciate the inherent strengths and limitations of the evidence base. As this is a narrative review, we did not apply a formal GRADE assessment, but we have clarified in the text that the current evidence remains largely derived from observational and retrospective studies, underlining the need for higher-quality data.
Use of temocillin in different patient population: Elderly sections mostly rely on retrospective data without statistical analysis. Elaborate the need for prospective, population-specific PK studies. Also, suggest caution in extrapolating dosing without pharmacometric validation in these groups.
We thank the reviewer for the suggestion. We have added a further clarification.
Safety and Tolerability: Add a table summarizing adverse events across studies with frequency and severity.
We thank the reviewer for this suggestion. However, we feel that an additional table would be redundant with the information already provided in the text and not fully aligned with the main objectives of this review.
Cost Analysis and Economic Considerations: Very narrow geographic scope (Sweden and Iran only) have been used. Assumptions underlying cost-savings are not critically analyzed.
We thank the reviewer for this helpful comment. We acknowledge that published economic evaluations of temocillin are currently limited to a few countries, notably Sweden and Iran. As we already note in the section, differences in healthcare costs, resistance patterns, and drug accessibility mean that the assumptions underlying cost-effectiveness analyses are highly context-dependent. To address the reviewer’s concern, we have further clarified this point in the text, underlining that broader and locally adapted economic evaluations are needed before general conclusions can be drawn.
Conclusion and Future Directions: The conclusion reiterates prior content and lacks a concrete future roadmap. Propose future research domains
We thank the reviewer for this useful comment. We have further expanded the discussion as suggested.